# Evaluation of the Margin of Stability during Gait Initiation in Young Healthy Adults, Elderly Healthy Adults and Patients with Parkinson’s Disease: A Comparison of Force Plate and Markerless Motion Capture Systems

**DOI:** 10.3390/s24113322

**Published:** 2024-05-23

**Authors:** Arnaud Simonet, Paul Fourcade, Florent Loete, Arnaud Delafontaine, Eric Yiou

**Affiliations:** 1LADAPT Loiret, Centre de Soins de Suite et de Réadaptation, 45200 Amilly, France; simonet.arnaud@ladapt.net; 2CIAMS, Université Paris-Saclay, 91405 Orsay, France; paul.fourcade@universite-paris-saclay.fr (P.F.); arnaud_94150@hotmail.fr (A.D.); 3CIAMS, Université d’Orléans, 45067 Orléans, France; 4Laboratoire GeePs—CENTRALESUPELEC, 91190 Gif-sur-Yvette, France; florent.loete@centralesupelec.fr; 5Laboratoire d’Anatomie Fonctionnelle, Faculté des Sciences de la Motricité, Université Libre de Bruxelles, CP 619-1070 Brussels, Belgium; 6Laboratoire d’Anatomie, de Biomécanique et d’Organogenèse, Faculté de Médecine, Université Libre de Bruxelles, CP 619-1070 Brussels, Belgium

**Keywords:** markerless motion capture, force plate, gait initiation, margin of stability, Bland–Altman, Bayes factor 01, Parkinson’s disease, healthy adults, biomechanics

## Abstract

Gait initiation (GI) is a functional task classically used in the literature to evaluate the capacity of individuals to maintain postural stability. Postural stability during GI can be evaluated through the “margin of stability” (MoS), a variable that is often computed from force plate recordings. The markerless motion capture system (MLS) is a recent innovative technology based on deep learning that has the potential to compute the MoS. This study tested the agreement between a force plate measurement system (FPS, gold standard) and an MLS to compute the MoS during GI. Healthy adults (young [YH] and elderly [EH]) and Parkinson’s disease patients (PD) performed GI series at spontaneous (SVC) and maximum velocity (MVC) on an FPS while being filmed by a MLS. Descriptive statistics revealed a significant effect of the group (YH vs. EH vs. PD) and velocity condition (SVC vs. MVC) on the MoS but failed to reveal any significant effect of the system (MLS vs. PFS) or interaction between factors. Bland–Altman plot analysis further showed that mean MoS biases were zero in all groups and velocity conditions, while the Bayes factor 01 indicated “moderate evidence” that both systems provided equivalent MoS. Trial-by-trial analysis of Bland–Altman plots, however, revealed that differences of >20% between the two systems did occur. Globally taken, these findings suggest that the two systems are similarly effective in detecting an effect of the group and velocity on the MoS. These findings may have important implications in both clinical and laboratory settings due to the ease of use of the MLS compared to the FPS.

## 1. Introduction

Postural stability is a prerequisite to the performance of most of our daily motor tasks. Gait initiation (GI), the transient period between quiet standing and steady-state walking, is a functional task classically used in the literature to evaluate the capacity of healthy individuals and neurological patients to maintain stability under dynamic conditions [1,2,3,4,5]. As stressed in the literature (e.g., [6]), the GI process is a particularly challenging task for patients affected with Parkinson’s disease (PD) because it combines motor and cognitive components of movement preparation (referred to as “anticipatory postural adjustments”) and movement execution (the step itself). Therefore, biomechanical indicators of stability are of utmost importance to evaluate the efficiency of the postural control system and to better understand the physiopathology of postural disorders in neurological patients [2,3,4,5].

According to Hof et al. [7], the condition for postural stability in dynamic situations holds that the position of the vertical projection of the centre of mass (CoM) plus its velocity multiplied by a factor dependent on the square root of the leg length should be within the base of support (BoS). This vector quantity is called the “extrapolated CoM position”. According to the authors, this definition suggests a variable called the “margin of stability” (MoS), which corresponds to the minimum distance from the extrapolated CoM to the boundaries of the BoS, as a measurement of stability.

Following the pioneering work of Hof et al. [7], postural stability during GI has been widely evaluated through the computation of this variable, as obtained from force plate recordings (e.g., [8,9,10]). Force plates allow the computation of CoM kinematics (more specifically, CoM velocity and position, which are two components of the MoS) during the brief period of GI via application of Newton’s second law (e.g., [11]). Yiou et al. [12] further showed that the BoS size during GI could also be accurately computed from the centre of pressure (CoP) trace obtained from force plate recordings. However, as stressed in a recent study [13], the use of force plates has several drawbacks which may limit their use to routinely investigate GI in laboratory and clinical conditions: (i) they require time-intensive data processing that can introduce errors [14,15,16,17,18,19], (ii) they require highly trained operators for data processing and (iii) they need to be long enough to record the entire GI process (and a fortiori to record the following steps), which may be an issue when subjects initiate gait at maximum velocity on a small force plate.

Markerless motion capture (MLS) is a recent and easy-to-use innovative technology that has shown potential in overcoming these drawbacks of force plates [13]. MLS is a technology that consists of automated two-dimensional motion capture based on markerless video. It is coupled with deep learning software that uses the synchronized video data to estimate human poses in three dimensions. The deep learning software is trained on digital images of over 500,000 people. Once the human pose has been estimated, the three-dimensional model can be obtained to analyze the spatio-temporal parameters, kinematics and kinetics of body segments and CoM [20,21,22,23,24,25,26,27,28,29]. Note that the advantages and drawbacks of the MLS vs. FPS, as well as the limitation of inertial measurement units, to investigate the biomechanical organization of GI have been discussed in [13].

Simonet et al. [13] showed that MLS provides a reliable measure of two classical CoM velocity-based variables of GI, namely the peak CoM anteroposterior velocity (an indicator of motor performance) and the so-called “braking index”, an indicator of the capacity to brake the fall of the CoM under the effect of gravity [2,3,4,5]. However, to date, the question of whether this system could also provide a reliable measure of the MoS during GI—which implies the computation of several variables (the BoS, the CoM velocity and position), thus amplifying the potential errors on the measure—remains to be elucidated. Such knowledge might be highly relevant in studies focusing on postural control during GI, since the use of this system is growing very rapidly in laboratory and clinical settings.

Thus, this study tested the agreement between the MLS and the force plate system (FPS, considered here as the “gold standard”) to estimate the MoS during GI in young and elderly healthy adults and in PD patients. In the present study, the MoS along the mediolateral (ML) direction was of special interest, because controlling stability along this direction is known to be particularly challenging during locomotor tasks (e.g., [5,30,31,32]).

Note that the present paper is the companion paper of [13]. The same populations and protocol were used in these two papers and will be described briefly below.

## 2. Materials and Methods

### 2.1. Participants

Thirty-three participants divided into three groups (young healthy [YH], elderly healthy [EH] and patients with Parkinson’s disease [PD]) took part in the experiment (see Table 1 for anthropometrical features). For the patients with Parkinson’s disease, the average elapsed time since diagnosis was 7.4 ± 2.8 years. Their score on the Hoehn and Yahr Scale was 2.0 ± 0.6 points. Their scores on the Unified Parkinson Disease Rating Scale–III and Montreal Cognitive Assessment were 33.8 ± 11.7 and 25.7 ± 2.3, respectively. Their mini-mental scores were 27.0 ± 1.7. All participants provided written consent after being informed of the nature and purpose of the experiment, which was conducted in accordance with the Declaration of Helsinki and approved by the “Comité de Protection des Personnes Ile-de-France XI” under identification number 19028-60429.

### 2.2. Experimental Set-Up, Tasks and Conditions

Participants initiated gait from a standing posture and continued walking to the end of a six-meter track. Gait was initiated at a spontaneous velocity (“spontaneous velocity condition”, SVC) and at a maximum velocity (“maximum velocity condition”, MVC). Series of five GI trials were performed in each of these velocity conditions. The participants were barefoot and began walking when they felt ready after hearing an auditory signal.

The FPS was composed of two force plates (400 × 600 mm, AMTI, Watertown, MA, USA) placed in series and embedded at the beginning of the walking track (e.g., [8,13]). The participants initially stood on the first force plate and stepped forward onto the second force plate (Figure 1). The MLS was composed of twelve Qualisys hybrid cameras (Qualisys, Göteborg, Sweden), which recorded body motion during the entire GI process and the subsequent steps. Data obtained from the cameras were then transferred to Theia software version 2021.2.0.1675 (Theia3D, Kingston, ON, Canada) for reconstruction of body kinematics.

### 2.3. Raw Data Processing

#### 2.3.1. Force Plate System

The mediolateral (ML) acceleration of the CoM was determined from the ML ground reaction force recorded by the force plates according to Newton’s second law. The ML CoM velocity and displacement were computed by successive numerical integrations of the corresponding ML CoM acceleration using the trapezoidal rule [8,9,10,11]. The calculations were performed with integration constant null, i.e., initial velocity and displacement equal to zero. The mediolateral CoP coordinate was calculated from force plate data in accordance with the manufacturer’s instructions (AMTI Manual).

#### 2.3.2. Markerless Motion Capture System

The MLS recorded the 3D full-body position. CoM velocity along the ML direction (y’COM(t)) was computed as follows:y’COM(t) = [yCOM(t) − yCOM(t − 1)] ∗ F(1)
where yCOM(t) and yCOM(t − 1) are the ML position of the CoM at time t and at the previous frame (t − 1), respectively.

The sampling frequency F was set at 85 Hz for both the MLS and the FPS. This frequency corresponds to the limit of the MLS in full HD mode. The same frequency was used for the FPS to allow comparison with the MLS. For both measurement systems, data were filtered using a no-lag low-pass Butterworth second-order filter with a 15 Hz cut-off frequency. Qualisys track manager software version 2021.2 was used to synchronize the signals from both measurement systems.

### 2.4. Margin of Stability

The concept of “margin of stability” (MoS) introduced by Hof et al. [7] was used to quantify ML stability at the time of swing-foot contact, corresponding to the time of maximum instability. The MoS was computed as the difference between the ML boundary of the BoS (BoSymax) and the ML position of the “extrapolated CoM” at this time (YcoMFC), i.e.,
MoS = BoSymax − YcoMFC(2)

YcoMFC was calculated as follows:YcoMFC = yCoMFC + y’CoMFC/ω0(3)
where yCoMFC and y’CoMFC are the ML CoM position and velocity at foot-contact, respectively; ω0 is the eigenfrequency of the body modelled as an inverted pendulum, calculated as
(4)$$\omega 0=\sqrt{\frac{g}{l}}$$
where *g* = 9.81 m/s^2^ is the gravitational acceleration and *l* is the length of the inverted pendulum which, in this study, corresponded to 57.5% of body height [33].

ML stability at foot contact is ensured on condition that YcoMHC is within the BoSymax, which corresponds to a positive MoS.

### 2.5. Experimental Variables

For each participant and each trial, the different components of the MoS, i.e., yCoMFC, y’CoMFC and BoSymax, were computed using both measurement systems. These two sets of values were used to calculate YcoMFC from Equation (3) and then the MoS from Equation (2). yCoMFC and y’CoMFC were taken directly from the kinematical traces obtained using the two measurement systems (Figure 2). BoSymax was computed from the MLS recordings as the ML distance between the two heels at the time of swing-foot contact [12]. As previously shown in the literature, the BoSymax could also be computed using the ML CoP trace obtained from the FPS recordings (e.g., [12]). As can be seen from Figure 2, the ML CoP trace reached two quasi-plateaus during GI, corresponding to the two successive single-stance phases. The difference between the maximal ML CoP values reached during these two phases corresponded to BoSymax.

### 2.6. Statistics

To investigate the agreement between the FPS and MLS, a Bland–Altman (BA) analysis was performed. BA plots were generated for each velocity condition and group, with the horizontal axis representing the average MoS obtained using the two measurement systems (i.e., [MoSFPS+MoSMLS/2]), and the vertical axis representing the difference between the two systems (i.e., [MoSFPS-MoSMLS]). The lack of agreement between the two measurement systems was summarized by calculating the bias, estimated by the mean difference (d) and the standard deviation of the differences (SD). The normality of the differences was checked using the Shapiro–Wilk test. The dispersion of the bias differences of each BA plot was quantified using 95% agreement limits. These limits correspond to 1.96 SD (upper limit) and −1.96 SD (lower limit). Absolute and relative values of biases and agreement limits were reported. Correlation analysis was also performed between values on the horizontal and vertical axes to test whether the biases changed according to the MoS values. The BA analysis was completed by descriptive statistics, which included MoS means and SD. Repeated measure (RM) ANOVAs were conducted on the MoS with the velocity condition (two levels: SVC vs. MVC) and the measurement system (two levels: MLS vs. FPS) as the within-subject factors, and the group (three levels: YH vs. EH vs. PD) as the between-subject factor. The alpha level for statistical difference was set at *p* = 0.05. Finally, the Bayes factor 01 was computed to contrast the following hypotheses: H0 (the null hypothesis, i.e., “the two measurement systems provide the same MoS”) vs. H1 (the alternative hypothesis, i.e., “the two measurement systems provide a different MoS”). It is generally admitted that if the Bayes factor 01 is above three, then the null hypothesis is validated [34,35,36].

## 3. Results

### 3.1. Description of the Biomechanical Traces

The time-course of the biomechanical traces obtained with the two measurement systems was very similar (Figure 2). The ML CoM velocity trace reached a peak value toward the stance leg side just before foot-off. This trace then dropped towards the swing-leg side to reach a second peak a few milliseconds after heel-contact. The ML CoM shift trace reached a peak value toward the stance leg side during the GI swing phase. It then fell to the swing leg side. The ML CoP trace reached two successive plateaus: the first corresponded to the first swing phase (from swing foot off to heel contact) and the second plateau corresponded to the second swing phase (from rear toe off to heel contact).

### 3.2. Bland–Altman Analysis

In SVC (Figure 3, left), BA analysis showed that 95% of the absolute differences between the two measurement systems ranged between −0.01 and 0.02 m for the YH group (which corresponded to a [−17 and 20%] range of relative difference), −0.02 and 0.02 m for the EH group ([−23 and 26%]) and −0.02 and 0.03 m for the PD group ([−28 and 30%]). In MVC (Figure 3, right), BA analysis showed that 95% of the absolute differences between the two measurement systems ranged between −0.02 and 0.03 m for the YH group ([−21 and 25%]), −0.02 and 0.03 m for the EH group ([−27 and 36%]) and −0.04 and 0.04 m for the PD group ([−41 and 37%]). As reported in the panels of Figure 3, absolute bias was very close to zero and the deviation was considered neglectable for each velocity condition and group, indicating that the accuracy of the MLS was good. In addition, there was no significant correlation between the values on the vertical axis ([MoSFPS-MoSMLS]) and the horizontal axis ([MoSFPS+MoSMLS/2]) in any BA plot.

### 3.3. Descriptive Statistics

RM ANOVA showed that there was a significant main effect of the group (F[2,476] = 5.7, *p* < 0.001) and the velocity conditions (F[1,476] = 28.1, *p* < 0.001) on the MoS (Figure 4). In contrast, there was no significant main effect of the measurement system (F[1,476] = 0.7, *p* = 0.41), nor any significant interaction between factors on this variable.

### 3.4. Bayes Factor 01

The Bayes factor 01 value showed that the null hypothesis (H0: “there is no difference of MoS between the two measurement systems”) was 7.4 times as likely as the alternative hypothesis (H1: “there is a difference of MoS between the two measurement systems”), which corresponds to “moderate evidence” (Figure 5).

## 4. Discussion

This study tested the agreement between the MLS and the FPS (considered as the gold standard) to estimate the MoS during GI in healthy adults (both young and elderly) and in PD patients. Agreement was investigated using the BA method, classical descriptive statistics and Bayes factor 01.

Visual analysis of Figure 2 showed that for both ML CoM velocity and ML CoM displacement, the time-course of the traces computed with the MLS and the FPS were almost superimposed. This was the case for all three groups and both velocity conditions. This first-instance observation suggests that the agreement between the two measurement systems for these MoS components was good (not reported in the present study), which further presumes good agreement for the MoS.

The results provided by the different statistics used in this study confirmed this first-instance observation. First, descriptive statistics showed that there was no significant effect of the measurement system on the MoS. These MoS values reached around 8 cm, which is slightly above values reported in several previous papers on GI (around 5–6 cm, e.g., [8,10]), but in agreement with Delafontaine et al. [9] (around 8 cm). Importantly, there was a significant main effect of the group and the velocity condition on the MoS, without any significant two-by-two or triple interaction between the group, the velocity condition and the measurement system. In other words, the two measurement systems were equally sensitive in detecting an effect of the group and the velocity condition on the MoS. Note that our previous study using the same set of participants and the same experimental conditions as in the present study showed that GI motor performance (in terms of peak anteroposterior velocity along the progression axis) increased significantly from SVC to MVC [13]. Increasing GI velocity from SVC to MVC resulted in an increase of the MoS, which was detected similarly by the two measurement systems. Second, the BA plots showed that the mean MoS biases were virtually zero for both velocity conditions and for all groups, indicating that the accuracy of the MLS was good. Finally, the Bayes factor 01 reached a value of 7.4, corresponding to “moderate evidence” in favor of the null hypothesis [34,35,36], i.e., “both measurement systems provide the same MoS”. Note that Bayes factor 01 offers an advantage over the *p*-values provided by descriptive statistics because it quantifies the evidence for and against two competing hypotheses, which cannot be achieved using *p*-values.

It is noteworthy that the upper relative limit of agreement of the MoS reached 20%, 26% and 30% in the YH, EH and PD groups in SVC, respectively, and 25%, 36% and 42% in the YH, EH and PD groups in MVC, respectively. Such relatively high values (i.e., values > 10% [12,13]) were expected since the MoS resulted from a difference between two quantities, the BoSmax and the extrapolated CoM (these quantities reached around 14 cm and 5 cm, respectively, all groups and velocity conditions combined). Therefore, for each trial, the between-systems difference on the MoS was equal to the algebraic sum of the between-systems difference on each of these two quantities. Thus, the small relative differences on each of these quantities (around 10%, all groups and velocity conditions combined; not reported in the present study) resulted in large relative differences on the MoS. As reported above, these large relative differences increased from the healthy young adults to the elderly healthy and PD patients. These findings call into question the agreement between the MLS and FPS to measure the MoS on a trial-by-trial basis.

Other markerless systems to the one used in the present study are currently available on the market, and they can be used to reconstruct human kinematics. For example, RGB and RGB-D cameras have proven to be efficient for real-time estimation of whole-body poses in interactive systems and games [37,38]. However, the Sun’s infrared range can cause interferences with the signal, which is not the case for the system used in the present study. Because of this constraint, the use of RGB and RGB-D cameras require that experiments be carried out under studio conditions to avoid daylight interferences, which is not practical. Phone cameras with open source software are very easy-to-use tools, but data obtained from this system are as yet not sufficiently accurate for research settings like the present one. A markerless prenium system was chosen in the present study; it has been shown to provide data on lower limbs gait kinematics that are as reliable as those from marker-based systems, as revealed by the inter-session variability, inter-trial variability and inter-session variability ratios [24]. Estimates from this markerless motion capture system were also very similar to those obtained from marker-based motion capture in terms of ankle and knee joint angles and moments [23,29]. This system only requires calibration of the area captured by the cameras. Finally, in all the studies cited above, Theia3D deep-learning software was used. Riazati et al. [39] have shown that under uncontrolled clothing conditions, data derived from Theia3D are associated with acceptable levels of absolute test-retest reliability.

## 5. Conclusions

In conclusion, the results of this study suggest that, although relatively high differences in the MoS do occur between MLS and FPS on a trial-by-trial basis, these two measurement systems are similarly effective in detecting an effect of the group and velocity on the MoS. Considering the population as a whole, these results further suggest that MLS and FPS provided equivalent MoS values. Thus, these findings add to the growing body of evidence in the literature [13] that suggests MLS as a sufficiently reliable tool to investigate biomechanical GI features in healthy participants and PD patients. This statement may have important implications in both clinical and laboratory settings due to the ease of use of MLS compared to FPS.

## Figures and Tables

**Figure 1 sensors-24-03322-f001:**
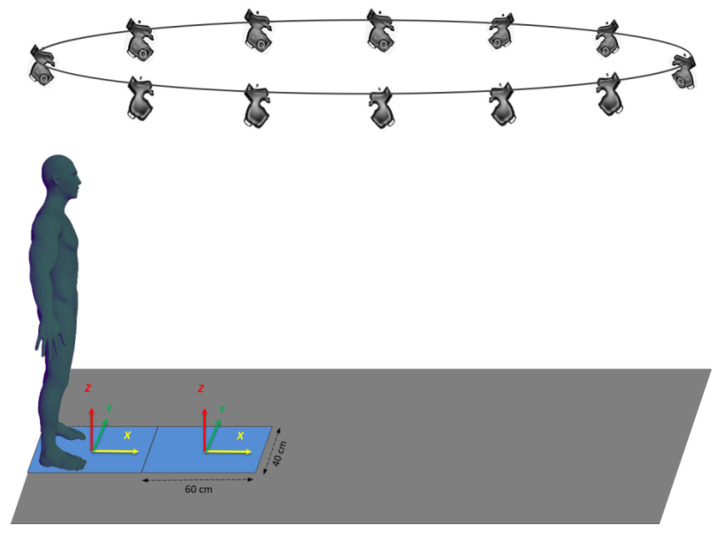
Experimental set-up showing one participant’s initial posture and the positions of the force plates and markerless cameras.

**Figure 2 sensors-24-03322-f002:**
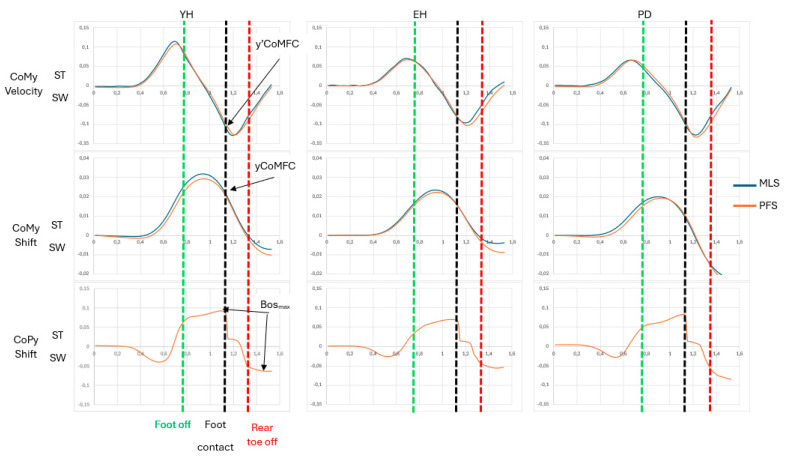
Typical biomechanical traces of the centre of mass (CoMy) velocity, CoMy shift and centre of pressure (CoPy) shift along the mediolateral direction obtained using the force plate system (FPS, red trace) and the markerless motion capture system (MLS, blue trace) in the three groups and two velocity conditions. The mean traces of the five trials obtained in the maximum velocity condition are reported for a representative participant of the young healthy adults (YH), elderly healthy adults (EH) and Parkinson’s disease patients (PD). CoM velocity and CoM/CoP shift (ordinate) are expressed in meters/second and meters, respectively. Time (abscissa) is expressed in seconds. ST and SW indicate stance limb and swing limb, respectively. yCoMFC, y’CoMFC: mediolateral position and velocity of the CoM at foot-contact. BoSymax: mediolateral base of support size. A positive variation of the traces indicates a displacement or velocity toward the stance leg side. A negative variation of the traces indicates a displacement or velocity toward the swing leg side.

**Figure 3 sensors-24-03322-f003:**
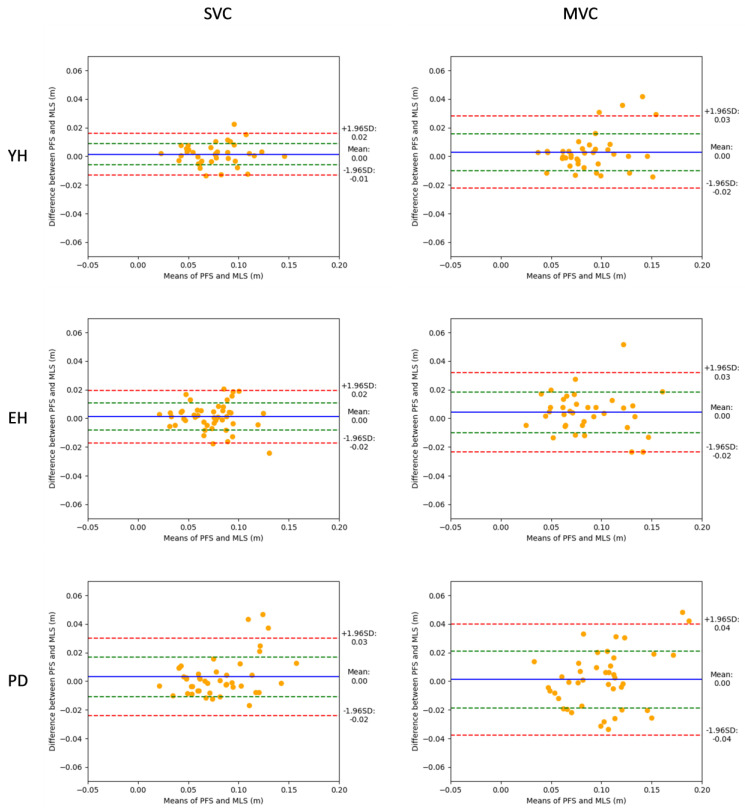
Bland–Altman plots showing the margin of stability values obtained with the two measurement systems (mean values, in abscissa), against the difference between these two systems (in ordinates), for the young healthy adult (YH), elderly healthy adult (EH) and PD groups. Each point represents one trial for one participant. The absolute values obtained in the spontaneous (**left**) and maximum (**right**) velocity conditions (SVC and MVC, respectively) are reported. For each plot, the 95% limits of agreement (red dotted lines), standard deviation (green dotted line) and bias (blue full line) are shown.

**Figure 4 sensors-24-03322-f004:**
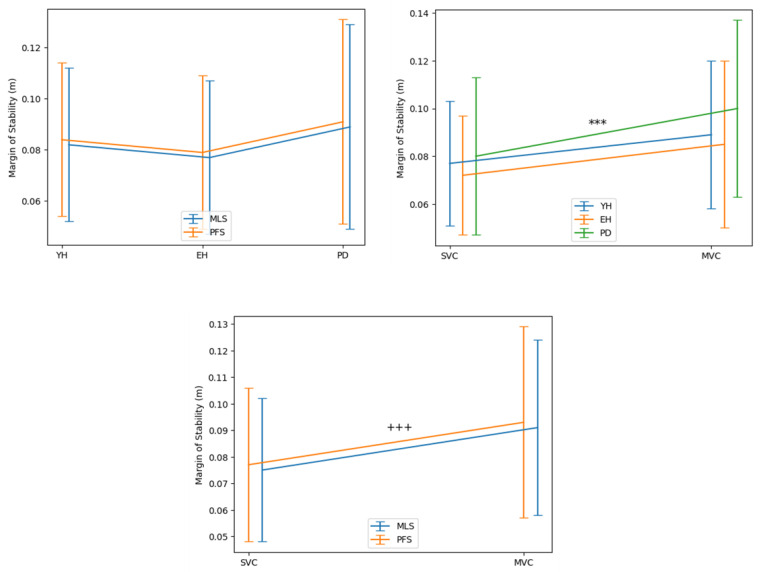
Main effect of the measurement system, the group, and the velocity condition on the MoS. FPS, MLS: force plate system and markerless motion system, respectively. SVC, MVC: spontaneous and maximum velocity condition, respectively. YH, EH, PD: young healthy adults, elderly healthy adults and PD group, respectively. Mean values (all participants combined) ± 1 standard deviation are reported. Middle panel. ***: main effect of the group with *p* < 0.001. Lower panel. +++: main effect of the velocity condition with *p* < 0.001.

**Figure 5 sensors-24-03322-f005:**
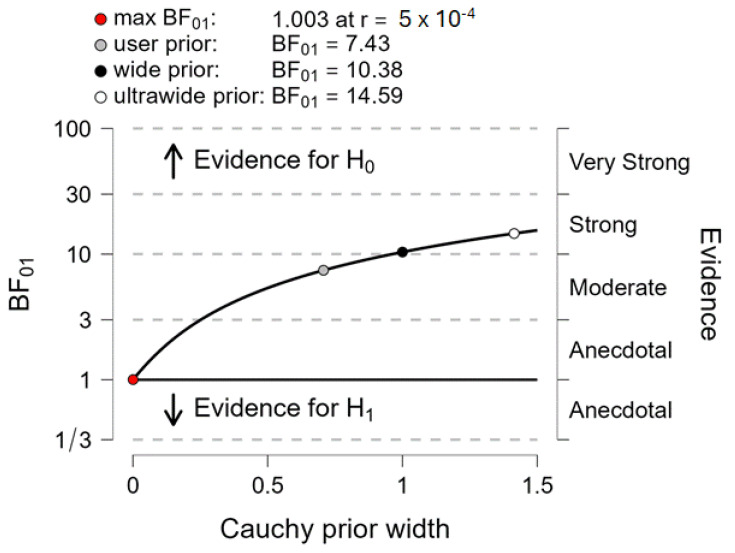
Between-systems comparison with the Bayes factor 01 applied to the margin of stability. The plot shows the Cauchy prior width (abscissa) vs. Bayes factor value (ordinate). Reported in the plot are the user value (grey dot), wide value (black dot) and ultrawide value (white dot). H0: null hypothesis (“there is no difference between the two systems”); H1: alternative hypothesis (“there is a difference between the two systems”). The graph shows the BF01 for user-specified a prior (r = 1/√2), wide a prior (r = 1) and ultra-wide a prior (r = √2). The maximum BF01 is achieved when the prior width r is set to 5 × 10^−4^. The evidence for the alternative hypothesis is relatively stable over a wide range of a priori distributions, suggesting that the analysis is robust. The evidence in favor of the H0 hypothesis is moderate.

**Table 1 sensors-24-03322-t001:** Participants’ anthropometrical features. BMI: body mass index. There was no significant effect of the group on the body mass, height, shoe size, body mass and BMI.

Group	Age (Years)	Gender (Female/Male)	Body Mass (kg)	Height (m)	BMI (kg/m^2^)	Shoe Size (EU)
PD (n = 12)	68.4 ± 5.1	1/10	70.5 ± 11.8	1.70 ± 0.07	24.5 ± 3.6	41.3 ± 1.5
YH (n = 10)	24.7 ± 0.7	4/5	70.2 ± 13.5	1.70 ± 0.12	24.7 ± 0.7	40.3 ± 3.7
EH (n = 11)	66.5 ± 3.6	8/3	63.9 ± 10.1	1.65 ± 0.06	23.4 ± 3.4	39.3 ± 1.8

## Data Availability

Data can be made available upon request to the first author of this study.

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
