# Peer review of "Evaluation of the Margin of Stability during Gait Initiation in Young Healthy Adults, Elderly Healthy Adults and Patients with Parkinson’s Disease: A Comparison of Force Plate and Markerless Motion Capture Systems"

_sensors, 2024, doi:10.3390/s24113322_

Round 1

Reviewer 1 Report

Comments and Suggestions for Authors

The work is very good, methodologically well constructed.  The only recommendation is to bring a slightly more detailed discussion about the statistical independence (A-nova) of the 5 levels in Figure 5.

It would be interesting to add a figure with histograms of the five levels showing that they have such averages that they are perceptible and different from each other.

Finally, from this analysis, improve the discussion and conclusions a little.

Author Response

Reviewer #1

The work is very good, methodologically well constructed.  The only recommendation is to bring a slightly more detailed discussion about the statistical independence (A-nova) of the 5 levels in Figure 5. It would be interesting to add a figure with histograms of the five levels showing that they have such averages that they are perceptible and different from each other. Finally, from this analysis, improve the discussion and conclusions a little.

Reply. We greatly thank the reviewer for his/her positive appreciation regarding of our work. The comment was however was not clear to us. The reviewer indeed proposed to improve the manuscript with a discussion about “the statistical independence (A-nova) of the 5 levels in Figure 5”. This figure corresponds to the between-systems comparison with the Bayes factor 01 applied to the margin of stability (MoS). It was not clear to us what are the “five levels” the reviewer refers to. The most important point we wished to stress was that the Bayes factor reached the value of 7.43, which indicated “moderate evidence” that both systems provided equivalent MoS. The caption of the figure 5 was further improved to be clearer on the meaning of the different values reported in the figure.

Also, the reviewer refers in his/her comment to ANOVA statistics. Repeated measures (RM) were indeed applied to the MoS in the present study. However, here again, it was not clear to us to what “five levels” of the ANOVA the reviewer may refer to (if so). RM ANOVAs were conducted on the MoS with the velocity condition (two levels: SVC vs MVC) and the measurement system (two levels: MLS vs. FPS) as the within-subject factors, and the group (three levels: YH vs. EH vs. PD) as the between-subject factor. So, there was three factors, and a total of seven levels. The main effects of these factors on the MoS (along with the report of the MOS values) are reported in the figure 4 (and not the figure 5).

Reviewer 2 Report

Comments and Suggestions for Authors

The authors present a study on the estimation of the margin of stability (MoS) measured at gait initiation (GI) with two technologies, i.e. a force plate system, which is nowadays the golden standard, and a markerless motion tracking system.

The main contribution of this work is a comparison of two technologies for the computation of the MoS at GI. The authors present an in depth analysis of the results that they obtained on a composite sample of 33 participants. Results are not of easy interpretation, as it comes from the honest discussion provided by the authors, with a moderate support towards equality of performance between the two systems.

This study is closely related to [13] from the same authors. Probably merging the two works in one could have been possible in the general framework of motor performance assessment.

The introduction is successful to make the reader understand the importance of evaluating stability at GI, and the fact that FPS are the current gold standard. In my opinion the same introduction should explain much better waht MLS are. There are many ways of reconstructing human kinematics with ML systems, which include inertial sensors, RGB single camera, RGBD (with depth information) based on infrared, laser, or stereovision, multiple RGB cameras, and many algortihms (see Openpose) to reconstruct motion. These solutions differ in terms of price, complexity of the setup, and other features. Among these MLSs, the authors selected one that is nearly as expensive as FPS, requires accurate calibration and a processing pipeline that is comparable to FPS.This discussion is swept with lines 70-72 and a bunch of references ([20-29]) that have not been discussed.

The method section also need to be improved with some more details:

- how did the authors account from noise in the computed acceleration that is likely to cause drift in the double integration process?

- What the effect of lowpass filter (please not the typo at line 150, which is the order?) in the acceleration profile that is integrated? What the effect on the velocity? Please not that events tracked (e.g. foot contact to toe off) occurr in less than 0.2 s.

- Did you check the effect of approximation [33] (line 73) on the l parameter? Is there a way of estimating it from other anthropometric measurements?

Author Response

Reviewer #2

We greatly thank the reviewer for his/her comment on our work which helped us clarify some important points of the manuscript. Please find below the point by point reply to each comment.

The authors present a study on the estimation of the margin of stability (MoS) measured at gait initiation (GI) with two technologies, i.e. a force plate system, which is nowadays the golden standard, and a markerless motion tracking system.

1) The main contribution of this work is a comparison of two technologies for the computation of the MoS at GI. The authors present an in depth analysis of the results that they obtained on a composite sample of 33 participants. Results are not of easy interpretation, as it comes from the honest discussion provided by the authors, with a moderate support towards equality of performance between the two systems. This study is closely related to [13] from the same authors. Probably merging the two works in one could have been possible in the general framework of motor performance assessment.

Reply. The present study is closely related to [13] because it is a “companion paper”. This point was clearly indicated in our cover letter and our correspondence with the Editor (Ms Yvonne Chu) who agreed with this view. To further clarify this point, this status of companion paper is now explicitly stated in the introduction section of the present paper (please see line 93). We chose to split this work on MLS validation during gait initiation in two companion papers in order to avoid presenting a single paper that might be too long and less comprehensive to the reader. Note also that most authors in the literature focus either on the MoS or on the braking index to evaluate postural control.

2) The introduction is successful to make the reader understand the importance of evaluating stability at GI, and the fact that FPS are the current gold standard. In my opinion the same introduction should explain much better waht MLS are.

Reply. Thank you for your positive comment on the introduction. Precisions on the MLS are added in the introduction (see lines 69-78):

“Markerless motion capture (MLS) is a recent and easy-to-use innovative technology that has shown potential in overcoming these drawbacks of force plates [13]. MLS is a technology that consists of automated two-dimensional motion capture based on markerless video. It is coupled with deep learning software that uses the synchronized video data to estimate human pose in three dimensions. The deep learning software is trained on digital images of over 500,000 people. Once the human pose has been estimated, the three-dimensional model can be obtained to analyze the spatio-temporal parameters, kinematics and kinetics of body segments and CoM [20-29].”

3) There are many ways of reconstructing human kinematics with ML systems, which include inertial sensors, RGB single camera, RGBD (with depth information) based on infrared, laser, or stereovision, multiple RGB cameras, and many algortihms (see Openpose) to reconstruct motion. These solutions differ in terms of price, complexity of the setup, and other features. Among these MLSs, the authors selected one that is nearly as expensive as FPS, requires accurate calibration and a processing pipeline that is comparable to FPS.This discussion is swept with lines 70-72 and a bunch of references ([20-29]) that have not been discussed.

Reply. The pros and cons of the MLS used in the present study vs. the FPS, as well as the limitations of the IMU to estimate the CoM kinematics, have already been discussed in the companion paper. We therefore not wish to discuss again these specific points in the present manuscript. This was made clear in the introduction (see lines 76-78).

Now, following the comment of the reviewer, the use of other markerless systems to evaluate the biomechanical organization of gait initiation was discussed lines 341-358:

“Other markerless systems than the one used in the present study are currently available on the market, and they can be used to reconstruct human kinematics. For example, RGB and RGB-D cameras have proven to be efficient for real-time estimation of whole-body pose in interactive systems and games [37,38]. However, the sun's infrared range can cause interferences with the signal, which is not the case for the system used in the present study. Because of this constraint, the use of RGB and RGB-D cameras require that experiments be carried out under studio conditions to avoid daylight interferences, which is not practical. Phone cameras with open source software are very easy-to-use tools, but data obtained from this system are to date not sufficiently accurate for research settings like the present one. A markerless prenium system was chosen in the present study for the following reasons. This system has been shown to provide data on lower limbs gait kinematics as reliable as marker-based systems, as revealed with inter-session variability, inter-trial variability and inter-session variability ratio [24]. Estimates from this markerless motion capture system were also very similar to those obtained from marker-based motion capture in terms of ankle and knee joint angles and moments [23,29]. This system only requires calibration of the area captured by the cameras. Finally, in all the studies cited above, Theia3D deep-learning software was used. Riazati et al. [39] have shown that under uncontrolled clothing conditions, data derived from Theia3D are associated with acceptable levels of absolute test-retest reliability.”

4) How did the authors account from noise in the computed acceleration that is likely to cause drift in the double integration process?

Reply. Noise was removed from the acceleration signal with the no-lag low-pass Butterworth 2nd order filter with a 15 Hz cut-off frequency. This cut-off frequency has been previously determined in the literature from a residual analysis (Winter, 1990) and have been shown to contain 95 to 99% from the spectral power of the signals (Sinclair et al., 2013), which means that the useful information from these data was preserved after filtering. Following this filtering, the accelerometric signal was double integrated to obtain the COM displacement. This method is classically used in the literature and has been shown to be valid over the brief period of time intervals in this study (e.g. Maki and McIlroy 1999).

References

  • Maki BE, McIlroy WE. (1999). The control of foot placement during compensatory stepping reactions: does speed of response take precedence over stability? IEEE Trans Rehabil Eng. 1999 Mar;7(1):80-90. doi: 10.1109/86.750556.
  • Sinclair, J., Taylor, P. J., & Hobbs, S. J. (2013). Digital filtering of three-dimensional lower extremity kinematics: an assessment. Journal of human kinetics, 39, 25–36. doi:10.2478/hukin-2013-0065
  • Winter DA (1990). Biomechanics and Motor Control of Human Movement, 2nd ed. Wiley, New York.

5) What the effect of lowpass filter (please not the typo at line 150, which is the order?) in the acceleration profile that is integrated? What the effect on the velocity? Please not that events tracked (e.g. foot contact to toe off) occurr in less than 0.2 s.

Reply. Following the comment of the reviewer, the MOS was computed in each condition and group using a 10-Hz, a 15-Hz filter and no filter at all. Statistical analysis showed that there was no significant effect of the filter on this variable. These additional data can be made available to the reviewer if required.

A no-lag low-pass Butterworth 2nd order filter with a 15 Hz cut-off frequency was used. The typo was corrected.

6) Did you check the effect of approximation [33] (line 73) on the l parameter? Is there a way of estimating it from other anthropometric measurements?

Reply. As indicated in the methods, the l parameter of the formula (4) is the length of the body modeled as an inverted pendulum, i.e. it corresponds to the height of the body’s COM. The l value obtained i) with the Theia 3D's software and ii) with the formula [l = 57.5% * body height] (Winter 1990) was used in the formula (2) to compute the MOS from the MLS and PFS kinematics data, respectively (note that the l parameter can also be computed as the distance between the iliac spine and the heels). Now, the difference in the MOS value that can specifically be ascribed to a difference in the method of l computation is so small (i.e. around 2mm, which represents a difference of around 1%) that it can be considered as negligible. These additional data can also be made available to the reviewer if required.